# ABCD1 Transporter Deficiency Results in Altered Cholesterol Homeostasis

**DOI:** 10.3390/biom13091333

**Published:** 2023-08-31

**Authors:** Agnieszka Buda, Sonja Forss-Petter, Rong Hua, Yorrick Jaspers, Mark Lassnig, Petra Waidhofer-Söllner, Stephan Kemp, Peter Kim, Isabelle Weinhofer, Johannes Berger

**Affiliations:** 1Department of Pathobiology of the Nervous System, Center for Brain Research, Medical University of Vienna, 1090 Vienna, Austria; 2Program in Cell Biology, Hospital for Sick Children, Toronto, ON M5G 0A4, Canada; 3Department of Biochemistry, University of Toronto, Toronto, ON M5G 1A8, Canada; 4Laboratory Genetic Metabolic Diseases, Department of Clinical Chemistry, Amsterdam University Medical Centers, Amsterdam Neuroscience, Amsterdam Gastroenterology Endocrinology Metabolism, University of Amsterdam, 1105 AZ Amsterdam, The Netherlands; 5Division of Immune Receptors and T Cell Activation, Institute of Immunology, Center for Pathophysiology, Infectiology and Immunology, Medical University of Vienna, 1090 Vienna, Austria

**Keywords:** ABCD1, Abcd1 KO mice, cholesterol esters, cortisol, lipidomics, lipid droplets, lipid metabolism, VLCFA, X-linked adrenoleukodystrophy

## Abstract

X-linked adrenoleukodystrophy (X-ALD), the most common peroxisomal disorder, is caused by mutations in the peroxisomal transporter ABCD1, resulting in the accumulation of very long-chain fatty acids (VLCFA). Strongly affected cell types, such as oligodendrocytes, adrenocortical cells and macrophages, exhibit high cholesterol turnover. Here, we investigated how ABCD1 deficiency affects cholesterol metabolism in human X-ALD patient-derived fibroblasts and CNS tissues of Abcd1-deficient mice. Lipidome analyses revealed increased levels of cholesterol esters (CE), containing both saturated VLCFA and mono/polyunsaturated (V)LCFA. The elevated CE(26:0) and CE(26:1) levels remained unchanged in LXR agonist-treated Abcd1 KO mice despite reduced total C26:0. Under high-cholesterol loading, gene expression of SOAT1, converting cholesterol to CE and lipid droplet formation were increased in human X-ALD fibroblasts versus healthy control fibroblasts. However, the expression of NCEH1, catalysing CE hydrolysis and the cholesterol transporter ABCA1 and cholesterol efflux were also upregulated. Elevated Soat1 and Abca1 expression and lipid droplet content were confirmed in the spinal cord of X-ALD mice, where expression of the CNS cholesterol transporter Apoe was also elevated. The extent of peroxisome-lipid droplet co-localisation appeared low and was not impaired by ABCD1-deficiency in cholesterol-loaded primary fibroblasts. Finally, addressing steroidogenesis, progesterone-induced cortisol release was amplified in X-ALD fibroblasts. These results link VLCFA to cholesterol homeostasis and justify further consideration of therapeutic approaches towards reducing VLCFA and cholesterol levels in X-ALD.

## 1. Introduction

X-linked adrenoleukodystrophy (X-ALD, OMIM #300100) is a progressive neurometabolic disorder with a combined male/female incidence of 1 in 14,700 newborns [1,2]. The molecular cause of X-ALD is mutations in the ABCD1 gene encoding a peroxisomal membrane transporter, mediating the transport of CoA-activated, very long-chain fatty acids (VLCFAs, ≥C22:0) into the peroxisomal lumen. Although otherwise intact peroxisomes are present in X-ALD, this single-transporter deficiency impairs the import of VLCFAs into these organelles, preventing their subsequent degradation via β-oxidation. Given that peroxisomes are the only organelle capable of VLCFA degradation in mammals, these fatty acids become incorporated into various lipid classes and accumulate in the tissues and body fluids of X-ALD patients, which constitutes a biomarker for the diagnosis of X-ALD [3,4]. How elevated VLCFA levels contribute to disease pathophysiology is still unclear. VLCFAs are known to impact the physical characteristics of cellular membranes, such as their permeability and fluidity [5,6]. Recent observations indicate that the amount of cellular VLCFAs is linked to innate immune cell activation [7]. Moreover, elevated oxidative stress and lipoxidation resulting in oxidative damage of various cell components have been suggested as a disease mechanism [8]. X-ALD presents a spectrum of symptoms with two main clinical representations: cerebral X-ALD (CALD) and adrenomyeloneuropathy (AMN). CALD is characterised by rapid inflammatory demyelination in the cerebral white matter, with peak onset in childhood. AMN involves slowly progressing axonal degeneration, affecting the spinal cord and peripheral nerves, and usually becoming symptomatic in young adulthood. More than 70% of male X-ALD patients additionally suffer from adrenal insufficiency, requiring corticosteroid replacement [9,10,11]. Female patients (heterozygous carriers) often develop symptoms of a milder, AMN-like neurologic phenotype at a later age (>40 years) [12].

Already fifty years ago, crystalline needles, probably consisting of cholesterol-ester-enriched lipid inclusions, were discovered in steroid hormone-producing cells and macrophages of the adrenal cortex, testis and phagocytes (microglia/macrophages) in the demyelinating brain tissue of X-ALD patients [13,14]. These tissues are all associated with high cholesterol metabolism, suggesting that cholesterol may play a significant role in X-ALD pathology. Throughout the human body, the highest levels of ABCD1 protein are found in cells with a high cholesterol turnover [15]. Interestingly, in the Abcd1–deficient mouse model for X-ALD, we observed elevated plasma cholesterol corresponding to the levels of wildtype (WT) controls fed a high-cholesterol diet [16]. Of note, a recent observation including X-ALD patients revealed a significant positive correlation of total serum cholesterol and LDL-cholesterol with VLCFA levels [17]. In vitro, cholesterol depletion obtained by treatment with cholesterol-lowering statins or by prolonged culture in a lipid-deficient medium was found to reduce VLCFA accumulation in fibroblasts derived from X-ALD patients [18,19]. Statins were never approved as a therapy to lower VLCFAs in X-ALD patients, as clinical studies failed to show a consistent reduction in VLCFA levels other than that associated with the decrease in total plasma LDL cholesterol [20]. The vast majority of male X-ALD patients suffer from adrenocortical insufficiency, resulting in impaired synthesis of steroid hormones, including cortisol. In addition, other steroid hormone disturbances, such as testicular insufficiency or androgenic hair loss, are also common in men with AMN [10]. Finally, several studies have reported that potentially toxic cholesterol metabolites, including 25-hydroxycholesterol and 7-ketocholesterol, accumulate in fibroblasts and plasma of X-ALD patients [21,22].

Excess free cholesterol and other lipids are either directly exported from the cell by the cholesterol transporters ABCA1 and ABCG1, which are localised in the plasma membrane, or stored in the esterified form within intracellular lipid droplets (LDs). These organelles are present in virtually all cells and are dynamically synthesised or broken down in response to environmental signals or cellular needs. LDs consist of a neutral lipid core (mainly cholesterol esters and triglycerides), surrounded by a monolayer of phospholipids and a specific set of stably or transiently associated proteins [23,24]. The composition and morphology of LDs vary among cell types; for example, adipocyte-associated LDs used as energy storage depots are enlarged in size and enriched in triglycerides (TG), whereas LDs found in adrenocortical cells or Leydig cells of the testes are smaller, more numerous and primarily consist of cholesterol esters (CE) used for steroidogenesis [25]. While LDs were initially thought to primarily serve as a storage site for excess lipids, research over the past decade has revealed other important functions of LDs, including the storage of signalling precursors and fat-soluble vitamins, regulation of lipophagy, buffering excess of potentially toxic lipids and misfolded proteins, and prevention of oxidative and ER stress [26]. Moreover, numerous human diseases, including cancer and neurodegenerative disorders, but also normal ageing, have been associated with dysfunction in LD dynamics, further emphasising their role in the cellular stress response [27,28]. In order to perform their diverse functions efficiently, LDs communicate closely with other cellular organelles, and contact sites between LDs and virtually all other cellular organelles, including peroxisomes, have been identified [29,30]. Of note, Chang and co-workers identified that tethering LDs to peroxisomes involves ABCD1, the protein affected in X-ALD [31].

The aim of this study was to elucidate how the lack of ABCD1 function and the associated VLCFA accumulation affects cholesterol homeostasis, including LD formation and steroidogenesis, by using primary human skin fibroblasts from X-ALD patients and X-ALD mice as model systems.

## 2. Materials and Methods

### 2.1. Human Dermal Fibroblasts

Primary human fibroblasts were obtained from skin biopsies of male patients affected by X-ALD (n = 15) or controls (n = 6) with no metabolic disease. Table 1 provides a summary of the details concerning X-ALD fibroblast lines, encompassing ABCD1 mutations, ABCD1 protein modifications and relative amounts as well as the phenotypical manifestation observed in the patients. All studies involving human fibroblasts were approved by the Ethical Review Board of the Medical University of Vienna (application no. 729/2010). The fibroblasts were cultured in standard RPMI 1640 (Sigma Aldrich), supplemented with 1% Penicillin/Streptomycin, 1% glutamine and 1% Fungizone (all Invitrogen) and either 10% fetal bovine serum (FBS, Gibco Life Technologies), here termed “complete medium”, or 10% lipid-depleted FBS (Biowest), here termed “lipid-depleted medium (LDM)” at 5% CO_2_ and 37 °C. For lipidomic analysis, the fibroblasts were cultivated as described [32,33]. For the cholesterol treatment experiments, cells were first starved in LDM and then incubated for 24 h, 48 h or 5 days in LDM, supplemented with either 10 or 20 µg/mL cholesterol (Sigma Aldrich) dissolved in ethanol (EtOH) or vehicle (EtOH).

### 2.2. Mice

The mouse work involved adult, male Abcd1^+/y^ (WT) or Abcd1^−/y^ (KO) mice with the Abcd1^tm1Kan^/J allele [34] back-crossed to the C57BL/6J strain for over 20 generations. All mice for the study were bred and maintained at the in-house facility of the Medical University of Vienna in a temperature and humidity-controlled room on a 12:12 h light–dark cycle with free access to water and standard rodent chow (R/M-H, Ssniff^®^, Soest, Germany). Alternatively, at the age of 6.5 months, mice received R/M-H chow supplemented with 100 mg/kg TO901317 (Cayman Chemical, pressed into pellets by Ssniff) for 10 weeks [32]. Experimental cohorts were generated by mating heterozygous females and C57BL/6J males producing Abcd1 KO and WT male littermates, for which the genotype was obtained by PCR as described previously [35], and sacrificed for brain and spinal cord analysis at the age of 8 months (lipidomics) or 12 months (RT-qPCR and histology). All procedures were in compliance with Austrian regulations (BGBl. II Nr. 522/2012) and the European Union Directive 2010/63/EU for humane care and handling. The study procedures were approved by the local Animal Care and Use Committee of the Medical University of Vienna and by the Austrian Federal Ministry of Education, Science and Research (BMBWF-66.009/0174-V/3b/2019).

### 2.3. Lipidomics

Lipidomics was performed as previously described [36]. Brain and spinal cord tissue of 8-month-old Abcd1 KO and WT mice were dissected, immediately frozen in liquid nitrogen and stored at −80 °C. Tissue homogenates for lipid analysis were generated by adding water and a 5 mm stainless steel bead to the tissue, followed by homogenisation using a Qiagen TissueLyser II. The protein concentration of the tissue homogenates was determined using the Pierce bicinchoninic acid (BCA) protein assay. A sample amount equivalent to 1 mg of protein was used for lipid analysis. The data were normalised using internal reference standards.

### 2.4. Live Cell Imaging of Lipid Droplet (LD) Formation

LD induction was evaluated using the Incucyte^®^ SX5 Live-Cell Analysis System (Sartorius AG, Göttingen, Germany). Fibroblasts (n = 4 or n = 6) were seeded into 24-well plates and cultivated until ~80% confluency. Then, the cells were starved in LDM for 72 h before the addition of cholesterol and the LD fluorescent dye, BODIPY™ 493/503 (4,4-Difluoro-1,3,5,7,8-Pentamethyl-4-Bora-3a,4a-Diaza-s-Indacene, Invitrogen, final concentration 2 µg/mL), both in EtOH as solvent. LD induction (emitting green fluorescence, 503 nm) was recorded using an x20 objective and the green channel for the desired time. The results were analysed using Incucyte^®^ software, exported to GraphPad Prism v. 8 for visual display and further statistical analysis and expressed as green fluorescence area/cell area.

### 2.5. Evaluation of Cholesterol-Induced Cellular LD Content by Microscopy

Fibroblasts were seeded on glass coverslips in 12-well plates and cultivated in complete medium until ~80% confluency. After starvation in LDM for 72 h, fibroblasts were treated with 10 or 20 µg/mL cholesterol for the indicated time. After fixation with 4% or 2% paraformaldehyde (PFA), cells were stained with either Oil Red O (ORO, ICN Biomedicals Inc., Aurora, OH, USA) and haematoxylin or BODIPY™ 493/503 and DAPI as described elsewhere [37,38], and visualised using light microscope (Leica Reichert Polyvar 2, ×100 objective) and confocal microscopy (LSM700, Zeiss, Oberkochen, Germany, ×63 objective), respectively. The acquired images were evaluated using ImageJ 1.52a software, and the results were expressed as LD area/cell.

### 2.6. Evaluation of LD Content in the Spinal Cord of Abcd1 KO Mice

Male 12-month-old Abcd1 KO and WT mice (n = 7 each) were deeply anaesthetised with ketamine/xylazine before transcardial perfusion with phosphate-buffered saline (PBS, pH 7.4) followed by 4% PFA in 0.1 M Sörensen’s phosphate buffer. The spinal cord was dissected, post-fixed overnight in PFA, then cryopreserved in sucrose (15% overnight, followed by 30% for at least 48 h), all at 4 °C, before embedding in Tissue-Tek^®^ OCT freezing compound (VWR International) and freezing on dry ice. Frozen 12-μm thick spinal cord cross-sections were cut using a cryostat (Leica CM 3050 S). ORO staining was used to visualise neutral lipid/cholesterol ester accumulation. The stained sections were scanned using a Hamamatsu NanoZoomer 2.0 HT with an ×40 objective, and the digitalised views were exported to NDP.view2 software (accessed on 21 May 2019, https://www.hamamatsu.com) for analysis of the images. The genotypes were blinded and the number of LD-loaded motor neurons in the ventral horn of the lumbar spinal cord was counted manually and normalised to the ventral horn area. At least six sections per animal were analysed.

### 2.7. RNA Isolation and Reverse Transcription-Coupled Quantitative PCR (RT-qPCR)

RNA isolation from human fibroblasts was performed using the RNeasy mini kit (Qiagen, Hilden, Germany) according to the manufacturer’s instructions. Flash-frozen spinal cord of 12-month-old male Abcd1 KO and WT mice was homogenised using a rotor-stator Polytron^®^ (Kinematica AG, Malters, Switzerland). Total RNA was first extracted from the tissue homogenate in TRizol^®^ Reagent (Ambion, Kaufungen, Germany) and further purified on RNeasy Mini Kit columns (Qiagen) according to the manufacturers’ protocols. RNA (500 ng) was reverse transcribed using the iScript^TM^ cDNA synthesis kit (Bio-Rad, Hercules, CA, USA), and qPCR was performed using SYBR Green technology with SsoFast EvaGreen Supermix and the CFX96TM Real-Time PCR Detection System (Bio-Rad). Each cDNA sample was analysed in technical duplicates. Relative mRNA levels were calculated using the 2^−∆∆Cq^ method [39] and using HPRT1 as a reference gene. PCR primers are listed in Appendix A.

### 2.8. Cholesterol Efflux Measurement

Cellular cholesterol efflux was measured using a Cholesterol Efflux Assay Kit (Cell-based, Abcam, Cambridge, UK) according to the manufacturer’s instructions. Briefly, cells were cultivated in LDM supplemented with 10 µg/mL cholesterol for five days. Next, fluorescently labelled cholesterol was added in labelling medium for 1 h, followed by incubation in serum-free equilibration mix overnight, washed, and incubated with cholesterol acceptor (human serum) for 4 h. After the incubation steps, the supernatants and cell lysates (cells solubilised with cell lysis buffer) were transferred to separate microplates and analysed using a microplate reader. The cellular cholesterol efflux during 4 h was calculated as the ratio of the fluorescence intensity (RFU) in the supernatant and the total fluorescence intensity (RFU) of cell lysate + supernatant, and multiplied by 100 to express it as % cholesterol efflux.

### 2.9. Evaluation of Cortisol Production by Fibroblasts after Triggering with Progesterone

Fibroblasts were cultivated in 24-well plates with complete medium until they reached full confluency. Twenty-four hours before progesterone treatment, the medium was changed to the supplement-free FibroLife medium (Lifeline^®^ Cell Technology, San Diego, CA, USA). Next, cells were treated for 24 h with 1 µM progesterone (Sigma Aldrich/Merck, Darmstadt, Germany) in supplement-free FibroLife medium. Thereafter, the supernatants were collected for cortisol measurement, while the cells were collected in RIPA buffer for protein determination using a Pierce BCA protein assay kit (Thermo Scientific, Waltham, MA, USA). Cortisol was measured using Luminex XMAP^®^ bead array technology (MSHMAG-21K, Merck) according to the manufacturer’s instructions. The results were normalised to the cellular protein content.

### 2.10. Western Blot Analysis

Fibroblasts were lysed in RIPA buffer containing protease inhibitors (EDTA-free cOmplete tablets, Roche, Basel, Switzerland). Next, samples were mixed with 5× loading buffer, and proteins were separated on 10% polyacrylamide gel by SDS-PAGE. Separated proteins were transferred to a nitrocellulose membrane using semidry transfer (BioRad transfer system), followed by blocking with 4% non-fat milk powder. The following primary monoclonal antibodies were applied overnight at 4 °C to detect selected proteins: mouse anti-human ABCD1 (Euromedex ALD-1D6-AS, clone 2AL-1D6, 1:6000), mouse anti-human GAPDH (Proteintech, 1:50,000), mouse anti-human PLIN2 (Progen, 1:200), mouse anti-human β-actin (Chemicon, 1:500,000). The secondary antibody, goat anti-mouse conjugated with horseradish peroxidase (Dako, 1:20,000), was added for 1 h at room temperature. Chemiluminescent HRP Substrate (Immobilon^®^Western, Milipore, Burlington, MA, USA) was used for the detection of chemiluminescent signal using the Chemidoc XRS detection system (Bio-Rad). Quantitative analysis was completed using Image Lab 6.1 software (Bio-Rad).

### 2.11. Evaluation of Lipid Droplet–Peroxisome Contact Sites with Immunofluorescence Staining

Fibroblasts were seeded on coverslips before 72 h starvation in LDM followed by 72 h of supplementation with 20 µg/mL cholesterol treatment to induce LDs. For immunofluorescence staining, cells were fixed in 3% PFA, permeabilised with 0.1% Triton X-100 for 5 min and blocked with 2% FCS, 2% BSA and 0.2% fish gelatine in PBS for 30 min at room temperature, followed by PBS washing. For peroxisomal staining, cells were incubated overnight at 4 °C with a polyclonal rabbit anti-PMP70 antibody (1:1000, Invitrogen, Waltham, MA, USA), followed by incubation with secondary donkey anti-rabbit IgG Cy3 antibody (1:400, #IR715-165-150, Jackson, West Grove, PA, USA) for 1 h at room temperature. To visualise LDs, cells were incubated for 1 h with BODIPY™ 493/503 before mounting in Fluorescence Mounting Medium (Dako, Glostrup, Denmark). Confocal imaging of samples was performed using an LSM980 laser-scanning confocal microscope with an ×40 oil immersion objective (Plan-Apochromat ×40/1.3 oil DIC (UV) VIS-IR M27, ZEISS) and Zen software (3.3, blue edition). All images from the trial were taken using the same laser power. Super-resolution imaging of samples was performed using an LSM980 laser-scanning confocal microscope with the Airyscan 2 detector using an ×63 oil immersion objective (Plan-Apochromat ×63/1.40 oil DIC M27, ZEISS) and Zen software (3.3, blue edition). Raw super-resolution images were processed in Zen using the Airyscan processing method. The measurements of Manders’ co-localisation coefficients were performed using Volocity 6.3.1 software (PerkinElmer, Waltham, MA, USA). The same threshold was applied to all images from the same trial.

### 2.12. Statistical Analysis

For statistical analysis, one-way ANOVA or two-tailed unpaired Student’s *t*-test were used. Multiple testing was corrected using Sidak’s or Dunnett’s multiple comparisons test. *p*-values below 0.05 were regarded to indicate statistical significance. GraphPad Prism 8 (GraphPad Software) was used for statistical analysis and graphical representation of the data. The lipidomics results were analysed using Qlucore Omics Explorer 3.5 software and by performing the two-group comparison test. The same software was used to create heatmaps. The obtained *p*-values and log2-fold change values were extracted from Qlucore and imported into GraphPad for volcano plot preparation.

## 3. Results

### 3.1. Cholesterol Esters with Saturated VLCFAs as well as Mono and Polyunsaturated Fatty Acids Accumulate in X-ALD Fibroblasts and in the CNS of Abcd1 KO Mice

In X-ALD, the excessive VLCFAs accumulate in various lipid species; however, the most profound increase has been found within the CE fraction in the demyelinating areas of CALD brain tissue [40,41]. Thus, we were interested in whether CE-VLCFAs are also increased in human dermal X-ALD fibroblasts and in the non-demyelinating (Abcd1 KO) mouse model for X-ALD. We performed lipidomics to obtain the cholesterol ester-fatty acid (CE-FA) species profiles of primary skin fibroblasts derived from healthy controls and X-ALD patients and of brain and spinal cord tissue from 8-month-old WT and Abcd1 KO mice (Figure 1, Appendix A). We found significantly increased total amounts of CE-FAs in cultured X-ALD vs. control fibroblasts (Figure 1A). As expected, the most significantly enriched VLCFA in CE was C26:0 (Figure 1B). Next to the expected high amounts of saturated VLCFAs, we additionally found elevated levels of mono, di and polyunsaturated VLCFAs in cholesterol ester species in X-ALD vs. control fibroblasts (Figure 1B, Appendix A). It is worth noting that unsaturated LCFAs such as C16:3, C18:2, C18:4, C19:3, C22:1 and C22:2 were also enriched in CEs in X-ALD fibroblasts (Figure 1B, Appendix A). Moreover, lipidomic analysis of brain and spinal cord tissue of 8-month-old Abcd1 KO mice revealed a pattern similar to that in X-ALD fibroblasts (Appendix A). These mice were still pre-symptomatic, as consistent signs of neurodegeneration, characterised by spinal cord axonopathy but no cerebral involvement, emerge around 18–20 months of age [34,35,42]. Further comparison of the two tissues showed that both CE-(C26:0) and CE-(26:1) accumulate to a greater extent in the spinal cord than in the brain (Figure 1C,D), possibly reflecting the developing phenotype in aged Abcd1 KO mice. Together, these results demonstrate abnormal CE homeostasis in human fibroblasts and murine CNS tissue in X-ALD. It is worth noting that VLCFAs accumulate in various lipid species and to a significant degree also in triglycerides (TG). Our lipidomic analysis revealed the accumulation of TG(FA)s in X-ALD fibroblasts vs. control fibroblasts (Appendix A), with a trend that aligns with findings from other lipidomic studies on X-ALD fibroblasts [43] and spinal cord tissue from Abcd1 KO vs. WT mice [44]. We found a highly significant increase for TG(60:2), TG(62:2) and TG(62:3). Although we only have the sum of TG sn-1-3, based on fatty acid abundancy, all of them are highly likely to contain a VLCFA.

### 3.2. Imbalanced Cholesterol Homeostasis in X-ALD Leads to Increased Cholesterol Efflux

To elucidate how increased amounts of cholesterol esters impact cholesterol homeostasis in X-ALD, we first investigated the expression of key genes involved in cholesterol homeostasis (Figure 2A): 3-hydroxy-3-methylglutaryl-coenzyme A reductase (HMGCR), sterol-O-acyltransferase 1 (SOAT1), neutral cholesterol ester hydrolase 1 (NCEH1) and ATP-binding cassette transporter A1 (ABCA1) by RT-qPCR. Control and X-ALD fibroblasts were incubated for 5 days in RPMI medium containing standard FBS (complete medium), lipid-depleted FBS (LDM) or LDM supplemented with 10 µg/mL cholesterol (high-cholesterol).

The expression of HMGCR, the rate-limiting enzyme in the cholesterol synthesis pathway, is known to respond to these treatment conditions [45]. As expected, conditions of low cholesterol (LDM) induced a robust increase in HMGCR expression compared with complete medium (Control, *p* < 0.0001; X-ALD, *p* = 0.0023), whereas cholesterol loading (LDM + high-cholesterol) reduced the expression of HMGCR in both X-ALD (*p* = 0.0001) and control fibroblasts (*p* = 0.0050) (Appendix A). However, no differences in the extent of HMGCR regulation were observed between the genotypes (Figure 2B). Furthermore, the expression patterns of sterol regulatory element-binding protein 2 (SREBP2), encoding a key regulator of HMGCR and other genes related to cholesterol biosynthesis, along with the low-density lipoprotein receptor (LDLR), which is likewise regulated by SREBP2 [46], were strikingly similar to that observed for HMGCR (Appendix A).

SOAT1, also known as acyl-CoA:cholesterol acyltransferase 1 (ACAT1), is an enzyme localised on the ER membrane, where it catalyses the formation of fatty acyl-cholesterol esters for storage in LDs [47]. We found a trend for increased expression of SOAT1 in X-ALD compared with control cells already under standard conditions (*p* = 0.0820); and the difference was more pronounced (*p* = 0.0322) with cholesterol loading (Figure 2C). Whereas control cells were able to downregulate SOAT1 expression under long-term high-cholesterol load (*p* = 0.0029), the mRNA levels remained high in X-ALD cells (*p* = 0.5754) (Figure 2C, Appendix A).

NCEH1 catalyses the hydrolysis of CE-FAs at the ER [48]. Intriguingly, whereas the mRNA expression of NCEH1 showed a similar trend as observed in the case of SOAT1, the effect of cholesterol loading was not statistically significant (Control, *p* = 0.3147; X-ALD, *p* = 0.8286) (Appendix A). The group means were higher in X-ALD in all conditions, but the difference to control fibroblasts reached statistical significance only in complete medium (*p* = 0.0415) (Figure 2D). However, including the cholesterol metabolite 25-hydroxycholesterol (25HC) during cholesterol loading resulted in a more pronounced increase in NCEH1 mRNA expression in X-ALD cells, and the difference to control cells reached high statistical significance (*p* = 0.0073) (Appendix A).

Finally, we measured the expression of ABCA1, encoding a key player in reverse cholesterol transport. ABCA1 is localised in the plasma membrane, where it facilitates the export of free intracellular cholesterol to an extracellular acceptor, Apolipoprotein A1 (ApoA1), preventing the accumulation of toxic cholesterol [49,50]. The expression of ABCA1 mRNA was strongly upregulated in both control and X-ALD cells upon cholesterol-loading conditions (Control, *p* = 0.0005; X-ALD *p* = 0.0020), which is in accordance with the role of ABCA1 in cholesterol export. However, the upregulation of ABCA1 was markedly higher (*p* < 0.001) in X-ALD cells (about 10-fold change) compared with controls (about 6-fold change) (Figure 2E). As ABCA1 expression is mainly controlled by the nuclear receptor liver X receptor (LXR), an important transcriptional regulator of cholesterol and fatty acid homeostasis, we hypothesised that the LXR response is amplified in X-ALD. Indeed, when we supplemented the cells with the potent natural LXR ligand, 25-hydroxycholesterol during cholesterol loading, the dysregulation of SOAT1, NCEH1 and ABCA1 in X-ALD cells was even more pronounced (Appendix A). Thus, we next asked whether the increase in ABCA1 expression is reflected by functional alterations in the export of cholesterol from X-ALD cells. Accordingly, we loaded X-ALD and healthy control cells with cholesterol for 5 days before adding fluorescently labelled cholesterol and human serum as a cholesterol acceptor. Fluorescence measurements of the supernatants revealed that significantly higher amounts of cholesterol were exported from X-ALD cells compared with the controls (Figure 2F).

In a previous study, dietary treatment of X-ALD mice with the LXR agonist TO901317 was able to reduce total levels of VLCFAs in the brain and spinal cord [32]. Therefore, we reanalysed the data from that study to determine whether CE(26:0) and CE(26:1) were also reduced. Neither the levels of CE(26:0) nor CE(26:1) were altered after TO treatment (Appendix A).

Together, these data indicate that in X-ALD, the response to cholesterol exposure is altered, probably to counteract further increases in toxic free cholesterol as well as CEs, which may accumulate due to reduced CE-VLCFA hydrolysis.

### 3.3. Lipid Droplet Formation in Response to Cholesterol Treatment Is Increased in X-ALD Fibroblasts

Because excess intracellular cholesterol is stored in LDs as CEs and the expression of the cholesterol-esterifying enzyme is elevated in X-ALD, we analysed the dynamics of cholesterol-dependent LD induction in X-ALD cells. Control or X-ALD fibroblasts were first lipid-starved for 72 h in LDM to eliminate residual LDs, before incubation in LDM supplemented with cholesterol and the neutral lipid stain BODIPY™ 493/503. The formation of LDs was recorded continuously for up to 5 days using an Incucyte^®^ live imaging system. The analysis revealed significantly increased LD content in X-ALD cells compared with the healthy controls at all time points from 36 h onwards, with more pronounced differences over time (Figure 3A, Appendix A). We further confirmed our live cell-imaging results by confocal and light microscopy using BODIPY™ 493/503 (Figure 3B, Appendix A) or Oil Red O (ORO) (Figure 3C), respectively, to visualise LDs. Perilipin 2 (PLIN2), a ubiquitously expressed LD-associated protein, is a major component of the LD surface, where it plays an important role in regulating lipid metabolism. As the expression of PLIN2 is known to correlate with the cellular LD content, PLIN2 mRNA and protein levels have previously been used as markers for relative quantification of LDs [51,52]. Thus, we finally measured PLIN2 mRNA and protein levels in X-ALD and healthy control cells exposed to high-cholesterol. We found that the relative levels of both PLIN2 mRNA (Figure 3D) and PLIN2 protein (Figure 3E, Appendix A) were significantly higher in X-ALD cells compared with controls. In summary, these data consistently show that LD formation is increased in X-ALD fibroblasts exposed to cholesterol.

### 3.4. The Interaction between Peroxisomes and Lipid Droplets Is Low under Cholesterol-Loading and Is Not Affected by ABCD1 Deficiency

Recently, ABCD1 was proposed to be part of a tethering complex involving M1 Spastin at LD–peroxisome contact sites [31]. With ABCD1 expression shown to be upregulated by the loading of human macrophages with oxidised cholesterol metabolites [7], we next asked whether the lack or dysfunction of ABCD1 would affect the interaction between these two organelles under high-cholesterol conditions. Thus, we first analysed the ABCD1 protein level in different X-ALD primary fibroblast lines and calculated the amount relative to that of the control fibroblast line C1 (100% ABCD1) (Figure 4A, Table 1). Next, we loaded control and X-ALD fibroblasts, expressing different amounts of residual mutated ABCD1 protein with cholesterol for 72 h to induce LDs. Double immunofluorescence staining, using an antibody directed against the peroxisomal transporter ABCD3 (PMP70) to mark peroxisomes and BODIPY™ 493/503 to visualise LDs (Figure 4B), revealed a low extent of co-localisation between peroxisomes and LDs, which was not significantly different between control and X-ALD cells (Figure 4C). This observation suggests that increased LD formation under high-cholesterol loading is not linked to interactions between LDs and peroxisomes. In line with this finding, we did not observe any consistent differences between control and X-ALD cells, and neither the presence nor the functionality of ABCD1 protein affected the level of interaction between LDs and peroxisomes in X-ALD fibroblasts (Figure 4C,D).

### 3.5. Dysregulated Cholesterol-Related Gene Expression Is Reflected by Increased LD Accumulation in Neurons in the Spinal Cord of Abcd1 KO Mice

Abcd1 KO mice display key biochemical features of X-ALD, such as decreased β-oxidation and the accompanying elevated levels of VLCFAs—most prominently in the CNS and adrenal glands. Moreover, lipid inclusions, characteristic of X-ALD, have been found in the adrenal cortex of Abcd1 KO mice, but not in the CNS. Still, X-ALD mice do not show signs of either CALD or adrenal insufficiency [34,53]. However, Abcd1 KO mice develop late-onset axonopathy and locomotor impairment with concomitant oxidative stress and oxidative damage in the spinal cord motor neurons [8,42]. In good agreement, our lipidomic analysis revealed increased accumulation of CE-FAs in the spinal cord and brain of Abcd1 KO mice compared with the controls, but more extensive accumulation was observed in the spinal cord than in the brain (Figure 1D). In order to assess LD accumulation in ABCD1 deficiency in vivo, we analysed ORO-stained spinal cord from 12-month-old control and Abcd1 KO mice. Here, we focused on motor neurons in the ventral horn of the lumbar part of the spinal cord of WT and X-ALD mice. LDs were present in the motor neurons of both genotypes at this age; however, Abcd1 KO mice exhibited a significantly higher number of LD-laden cells than the WT controls (Figure 5A,B). It is important to acknowledge that since TGs also accumulate in the spinal cord of X-ALD mice [44], we could not determine the extent of CE and TG in the LDs. To investigate cholesterol homeostasis in the spinal cord of Abcd1 KO mice in more detail, we analysed the expression of the cholesterol-related genes Hmgcr, Soat1, Abca1 and Apoe in the spinal cord of 12-month-old WT and Abcd1 KO mice (Figure 5C). Similarly to our findings in human X-ALD fibroblasts, RT-qPCR revealed no genotype difference in the mRNA levels of Hmgcr. In contrast, Soat1, Abca1 and Apoe expression was significantly higher in the KO animals, reinforcing the results obtained in vitro from the X-ALD fibroblasts (Figure 2C,E). These results confirm that the elevated CE-FA levels and the altered homeostasis of cholesterol metabolism also promote LD formation in the X-ALD mouse model.

### 3.6. X-ALD Fibroblasts Produce a Higher Amount of Cortisol after Progesterone Stimulation

In X-ALD, most male patients suffer from adrenal insufficiency and require corticosteroid replacement therapy. In steroidogenic tissues, CEs are stored in the form of LDs and, upon hormone stimulation, free cholesterol is mobilised as a substrate for de novo steroid hormone production [25]. Human skin is also a steroidogenic tissue, and human skin fibroblasts can synthesise cortisol in vitro [54]. We hypothesised that altered cholesterol homeostasis in X-ALD fibroblasts might “lock” cholesterol in CEs, thus reducing the amount of free cholesterol available for cortisol production. Progesterone, a steroid and cortisol precursor inhibiting cholesterol esterifiction [55], was reported to stimulate LD lipolysis in murine cervical epithelial cells [56] and cortisol secretion from human fibroblasts [54]. We first set out to confirm that progesterone triggers cholesterol-induced LD lipolysis in healthy control fibroblasts supplemented with cholesterol to induce LD formation. Indeed, upon stimulation with progesterone for 24 h, we observed decreased cellular LD content compared with vehicle-treated cells, confirming the lipolytic effect of progesterone on LDs (Appendix A). Next, we exposed cholesterol-loaded X-ALD and healthy control cells to progesterone for 24 h and measured the cortisol level in the supernatants. The amount of cortisol secreted from X-ALD cells was significantly higher compared with control cells (Figure 6), possibly reflecting the increased availability of free cholesterol in X-ALD fibroblasts due to LD lipolysis and inhibition of CE formation. Of note, the induction of cortisol secretion upon progesterone treatment was independent of the upregulation of genes involved in cortisol synthesis, such as CYP11A1 and HSB11B, whose expression were unaltered (Appendix A).

## 4. Discussion

Cholesterol is an essential lipid for mammalian cells, and its homeostasis is precisely controlled in multiple regulatory pathways such as biosynthesis, uptake, efflux, storage, transport, utilisation and excretion. Disturbed cholesterol metabolism has been linked not only to atherosclerosis and cardiovascular disease but also to neurodegenerative disorders [57]. Although there are several indications of imbalanced cholesterol metabolism in X-ALD, little is known about its contribution to disease pathogenesis. Our study provides evidence on the role of disturbed cholesterol homeostasis in the pathophysiology of X-ALD. We found that total CE-FA levels are elevated in human X-ALD fibroblasts and in the CNS Abcd1 KO mice. Furthermore, lipidomic analyses of the CE fatty acyl-chain composition revealed increased incorporation of saturated VLCFAs and also of mono and polyunsaturated LCFAs and VLCFAs. In a lipidomic study conducted by Fourcade and co-workers on the spinal cord of 13-month-old Abcd1 KO mice, mice reported higher levels of total TGs, with a wide range of fatty acyl chains [44]. In that study, the total CE-FA levels showed a tendency to increase (1.5 times higher than the WT mean), but this difference was not statistically significant. It is worth noting that the analysis was performed on a small sample size of only three animals, limiting the ability for generalised conclusions [44]. Polyunsaturated fatty acids, which are prone to peroxidation, are a major player in the induction of ferroptosis, an iron-dependent form of regulated cell death [57,58]. Interestingly, ferroptosis has recently been associated with X-ALD and may play a role in the disease pathophysiology [59]. While acknowledging the importance of TG accumulation in the context of X-ALD, here we focused on investigating potential disruptions in cholesterol homeostasis stemming from the ABCD1 defect, which is of great importance for steroidogenic tissues or phagocytes removing myelin debris in CALD brain lesions.

Further, our lipidomic results revealed that the extent of CE-VLCFA content and neurodegeneration in the CNS of X-ALD mice are well matched. Not only were the levels in Abcd1 KO mice increased compared with WT controls, but in the spinal cord, already at an age preceding consistently increased axonopathy by about 12 months, both the CE(26:0) and CE(26:1) were markedly higher than in the brain, where signs of neuropathology are not featured, even in very old mice. Moreover, the boosted LD formation in the spinal cord of 1-year-old Abcd1 KO versus WT mice is well linked with the elevated CE-VLCFA levels. Our findings confirm recent observations on LD accumulation in the spinal cord of Abcd1/2 double-KO and Abcd1 KO mice [44,60]. In these studies, LD accumulation was attributed mainly to increased lipogenesis and elevated triglyceride levels due to mTOR/SREBP1c activation, which could be normalised by high-dose biotin treatment [44]. It is known that neurons can use lipids stored in LDs for cellular membrane production. However, chronically increased LD numbers in neurons have also been associated with cellular stress, ageing and neurodegenerative diseases such as Parkinson’s disease, multiple sclerosis or Alzheimer’s disease [28].

After challenging X-ALD fibroblasts in vitro with cholesterol, we observed an imbalance in the form of increased expression, or lack of repression, of the cholesterol-related genes, SOAT1, NCEH1 and ABCA1, compared with healthy controls. In contrast to these genes associated with the esterification state and the export of cholesterol, cholesterol synthesis, as reflected by HMGCR expression levels, was not differentially affected in X-ALD cells. Despite increased ABCA1-mediated cholesterol efflux, X-ALD cells still induced a higher amount of LDs than controls. Distinct acceptors are specific for different cholesterol efflux pathways. To measure specific cholesterol efflux, ApoA-1 is used to reflect ABCA1-dependent cholesterol efflux, while HDL is selective for ABCG1 or SR-B1-dependent cholesterol efflux. In our study, we used human serum as a cholesterol acceptor that contains a range of lipoproteins beyond ApoA-1, as well as albumin, which can bind cholesterol non-specifically and, thus, measure the combined transporter-specific and non-specific cholesterol efflux. We observed an analogous pattern with upregulated Soat1, Abca1 and Apoe but normal Hmgcr expression, associated with increased LD numbers, in the spinal cord of Abcd1 KO mice fed a standard diet. Within the CNS, APOE serves as the primary extracellular transporter of lipids and cholesterol, predominantly secreted by astrocytes. The APOE ε4 allele is the primary genetic risk factor for late-onset Alzheimer’s disease [61]. Furthermore, a recent observation revealed that male X-ALD patients with the APOE4 genotype exhibited increased cerebral involvement, as indicated by the severity assessed through MRI [62]. This makes our observation of Apoe upregulation in the spinal cord of Abcd1 KO mice intriguing, possibly suggesting an increased need for cholesterol export in X-ALD.

Furthermore, a recent whole-transcriptome analysis conducted on Abcd1-deficient BV-2 cells, a murine microglial cell model for X-ALD, revealed similar differences in cholesterol-related gene expression, with Soat1, Nceh1, Apoe and other genes related to the cholesterol efflux being significantly upregulated in mutant vs. WT BV-2 cells [63]. Nevertheless, also in this cell type, Raas and collaborators observed increases in Plin2 expression and LD abundance, despite the upregulation of cholesterol efflux genes. Intriguingly, BV-2 cells lacking Abcd1 displayed the accumulation of free cholesterol in the plasma membrane, as determined by flow cytometric analysis of Filipin III-binding kinetics. Generally, these differences were more pronounced in Abcd1/Abcd2 double-knockout BV-2 cells [63,64].

One reason for the accumulation of CE-FAs enriched in VLCFA species could be the fatty acyl-chain length preferences of the CE synthesising and hydrolysing enzymes. By using radioactive fatty acid substrates and post-mortem brain tissue, it was previously shown that the esterification rates in white and grey matter homogenates do not differ between X-ALD and controls, and the esterification rate for C26:0 was on average 25% of the rate towards C18:1 [65]. Another study investigating CE hydrolysis in post-mortem brain tissue from X-ALD patients found strongly reduced hydrolase activities against CE(24:0) and CE(26:0) compared with CE(18:1) [66]. In patient-derived fibroblasts, radioactively labelled CE(24:0) and CE(18:1) were both shown to be hydrolysed at comparable rates in X-ALD and control fibroblasts; however, the rate of hydrolysing CE(24:0) was only 5% of that for CE(18:1) [67]. Moroever, in rat brain homogenates, the esterifying activity towards VLCFAs was found to be relatively high (11% for C24:0 and 10% for C26:0 of the rate for C18:1) compared with the hydrolysis rates of 2% for CE(24:0) and 1% for CE(26:0) [65]. Ogino and Suzuki concluded that VLCFAs can be esterified in CE when present as free fatty acids but will not be readily hydrolysed, thus representing a conceivable biochemical mechanism for the accumulation of cholesterol esters with VLCFAs in the brain of X-ALD patients [65]. The advantage of these studies in brain or fibroblast homogenates is that the activities of all CE-synthesising and CE-hydrolysing enzymes are reflected including SOAT1 and NCEH1. Our observation that TO901317 treatment is able to reduce total VLCFA in the CNS [32] but not VLCFAs in the CE fraction (Appendix A), is in good agreement with the hypothesis that the hydrolysis of CE-VLCFA substantially contributes to VLCFA accumulation in the CE fraction.

Since oxysterols can arise from cholesterol through both enzymatic and non-enzymatic reactions, oxidative stress, another aspect of X-ALD pathology [68], could also play a role in the observed dysregulation of cholesterol homeostasis. Notably, oxysterols are strong activators of LXR, which induces the transcription of target genes involved in cholesterol export, such as ABCA1 and APOE [69], as well as the acyl-CoA desaturase SCD1, probably also contributing to our finding of increased levels of monounsaturated CE-FAs in X-ALD. Moreover, the Abcd1-deficient BV-2 microglial cell model exhibited elevated oxysterol levels [64]. The closely aligned gene expression findings of our in vitro and in vivo experiments further substantiate that the observed changes in cholesterol homeostasis constitute a consistent key finding that may play a role in the pathogenesis in X-ALD patients.

Addressing the potential effects of the imbalanced cholesterol metabolism on steroidogenesis, we further revealed that lipolysis and inhibition of CE formation by progesterone significantly stimulated cortisol secretion from X-ALD cells compared with healthy controls. While further research is needed to confirm that targeting LDs and CEs in the adrenals of X-ALD patients would positively impact cortisol production, our findings may stimulate new research strategies for a better understanding of the adrenal insufficiency in X-ALD patients.

In this study, our main focus was on LDs that are induced by cholesterol. Nevertheless, it is crucial to acknowledge that LDs exhibit substantial diversity. For instance, LDs induced by TGs in adipose tissue are considerably larger than those found in the adrenal cortex, and they also contain distinct proteins [52]. Our findings in cholesterol-loaded cells demonstrate only limited co-localisation between peroxisomes and LDs. Importantly, neither the presence nor the function of ABCD1 had any impact on this co-localisation. Given that fasting-induced lipolysis, triggered by activating adipose triglyceride lipase (ATGL), can increase LD–peroxisome interaction [70], it is plausible that analogous mechanisms might come into play concerning LDs containing cholesteryl esters (CE). Consequently, conducting an assessment of LD–peroxisome interactions under lipid-depleted conditions, after inducing LDs with cholesterol, holds significant potential and deserves further investigation.

Proper management of cholesterol levels is vital in the context of cerebral demyelination, when a large amount of myelin debris must be cleared by resident microglia and macrophages infiltrating the brain parenchyma to ensure resolution of inflammation and onset of regenerative processes [71]. We previously discovered that among the X-ALD-derived immune cells, monocytes/macrophages are metabolically the most severely affected and pro-inflammatory skewed [72,73]. Moreover, whole-transcriptome analysis of X-ALD-derived macrophages recently revealed cholesterol metabolism as one of the top dysregulated pathways, and that the transcriptional changes were enriched for genes encoding membrane proteins [7]. Therefore, further studies on the cholesterol metabolism in macrophages/microglia hold promise to provide more insight into the pathobiology of X-ALD.

Previously, we observed that the basal plasma cholesterol levels in Abcd1-deficient mice are elevated to the extent seen in WT mice fed a high-cholesterol diet for 6 weeks and could not be further increased by dietary cholesterol [16]. In the present study, we show that cholesterol efflux is greater in X-ALD-derived vs. normal human fibroblasts under high-cholesterol conditions. However, under these conditions, expression of SOAT1, responsible for the CE formation, is downregulated in the normal controls, whereas, in the X-ALD fibroblasts, the enzyme remains induced, which is in agreement with the increased LD formation that we observed compared with the healthy controls. In addition, the mRNA levels for NCEH1, catalysing the CE hydrolysis required for efflux of free cholesterol, remained induced, and expression of the cholesterol transporter, ABCA1, was strongly increased in the X-ALD fibroblasts, supporting the increased cholesterol efflux. Thus, it is tempting to speculate that incorporation of excessive VLCFAs into the plasma membrane leads to enhanced cholesterol uptake, thereby accelerating the cholesterol metabolism under high-cholesterol conditions to the limit of this cell type. Similar findings were observed in the X-ALD mouse model and in part, in human X-ALD, either in this paper or reported previously. For example, it was recently reported that a positive correlation exists between the VLCFA and cholesterol levels in the serum of X-ALD patients [17].

Taken together, disturbed cholesterol homeostasis, particularly in cells types with high cholesterol metabolism (oligodendrocytes, macrophages, microglia, Leydig cells or adrenocortical cells), is likely present in X-ALD. Interestingly, these cell types all exhibit very low levels of ABCD2, another peroxisomal ABC transporter, which upon overexpression in vitro or in vivo can compensate for the lack of ABCD1 [74,75]. This further contributes to the specification of cell types affected by X-ALD pathology. Of note, murine macrophages have higher, potentially compensatory Abcd2 expression than human macrophages, protecting them from a more severe metabolic phenotype and at least partly contributing to the lack of inflammatory cerebral ALD in mice [72,76]. Not even after the induction of reversible demyelination in the corpus callosum by the toxin cuprizone, resulting in an increased cholesterol burden over several weeks, did Abcd1 KO mice exhibit any major differences in the extent of de- and re-myelination [77]. In this model of acute white matter damage, the main difference to WT controls was a moderately accelerated loss of mature oligodendrocytes and associated axonopathy in the corpus callosum. Further in this context, the question could be posed ferroptosis, promoted by lipid peroxidation, could be a contributing factor to increased vulnerability of Abcd1-deficient oligodendrocytes.

The reason why cholesterol-lowering drugs, such as lovastatin, could not protect X-ALD patients from initiation of CALD is probably that the slight reduction in VLCFA and the decrease in cholesterol in peripheral tissues does not affect the cholesterol burden that ABCD1-deficient microglia/macrophages face in the CNS after initiation of demyelination. This hypothesis is in agreement with the success of haematopoietic stem cell transplantation or gene therapy, in which ABCD1 of the monocyte/macrophage lineage is corrected, enabling these cells to cope even with high cholesterol load under demyelinating conditions. Based on these findings, lowering VLCFA and cholesterol levels still appears as an attractive disease modifying target in X-ALD.

## Figures and Tables

**Figure 1 biomolecules-13-01333-f001:**
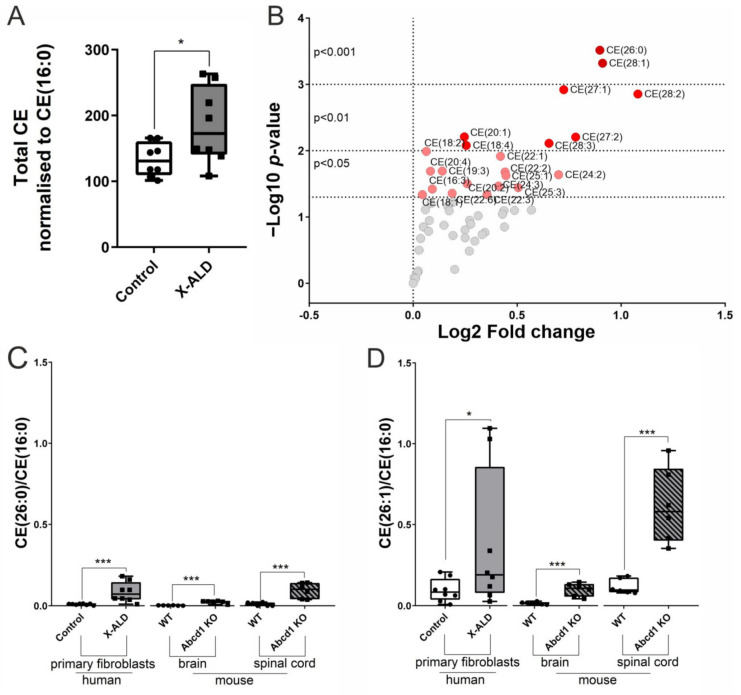
Cholesterol ester-fatty acid species are increased in primary human X-ALD fibroblasts and in the CNS of Abcd1 KO mice. (**A**–**D**) Lipidomic analyses were performed on control and X-ALD fibroblasts (n = 8 each) and on the brain and spinal cord of WT and Abcd1 KO mice (n = 6 each) using ultrahigh resolution liquid chromatography-mass spectrometry. The comparisons of total cellular cholesterol ester (CE) content (**A**) and CE-fatty acid composition (**B**) of control and X-ALD fibroblasts are displayed as a boxplot and volcano plot, respectively. The accumulation of CE-VLCFA species, represented by CE(26:0) in (**C**) and CE(26:1) in (**D**), was determined in human fibroblasts and murine CNS tissues and displayed as boxplots. All CE and CE(FA) levels were normalised to CE(16:0). Unpaired two-sided Student’s *t*-test was performed in (**A**). For (**B**–**D**), Qlucore Omics Explorer 3.5 software was used to analyse the lipidomic data statistically by conducting a two-group comparison test. The obtained *p*-values and log2-fold change values were used to create a volcano plot. The box plots (in (**A**,**C**,**D**)) show all the individual values and the median. * *p* < 0.05; *** *p* < 0.001.

**Figure 2 biomolecules-13-01333-f002:**
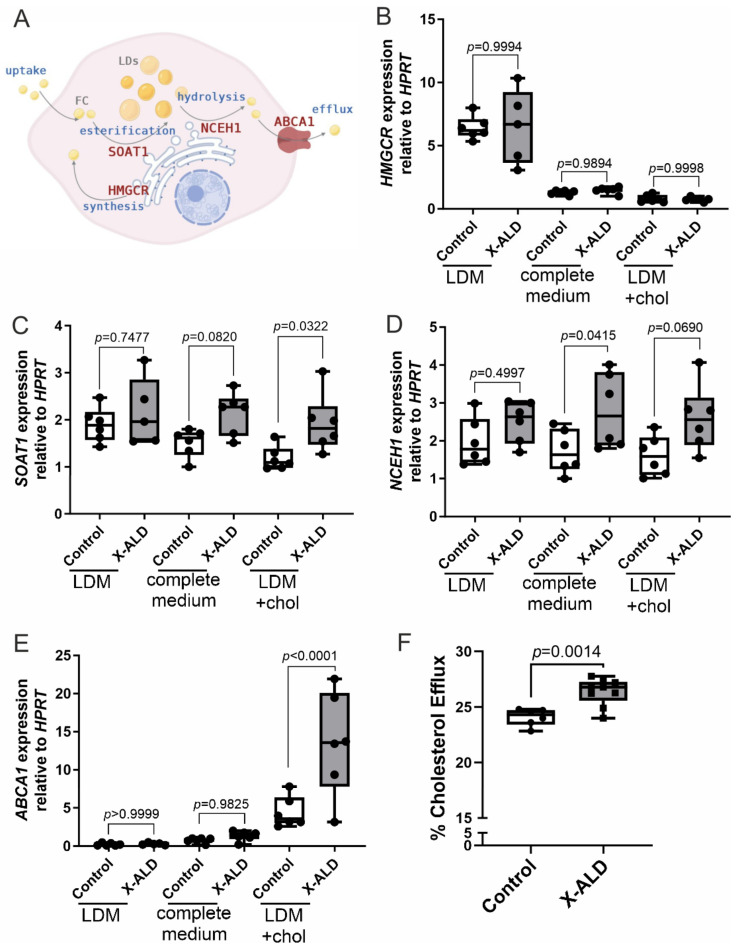
Imbalanced cholesterol homeostasis in X-ALD-derived fibroblasts leads to increased cholesterol efflux upon prolonged cholesterol loading. (**A**) Simplified schematic representation of cellular cholesterol turnover (FC, free cholesterol; LDs, lipid droplets). (**B**–**E**) Primary human control or X-ALD-derived fibroblasts (n = 6 each) were cultured for 5 days in lipid-depleted medium (LDM), RPMI with FBS (complete medium) or LDM supplemented with 10 μg/mL cholesterol (LDM + chol). RT-qPCR was carried out to determine the relative mRNA levels (normalised to HPRT1) of genes involved in cholesterol synthesis, HMGCR (**B**); cholesterol esterification, SOAT1 (**C**); cholesterol ester hydrolysis, NCEH1 (**D**); and cholesterol export, ABCA1 (**E**). (**F**) Cholesterol efflux was measured after 5 days of loading cholesterol in LDM + chol (control, n = 6; X-ALD, n = 9). The box plots in (**B**–**F**) show all individual values and the median. One-way ANOVA with Sidak’s multiple comparisons test was performed for statistical analysis of the data in (**B**–**E**). Unpaired two-sided Student’s *t*-test was performed in (**F**). The image in (**A**) was created using BioRender.com.

**Figure 3 biomolecules-13-01333-f003:**
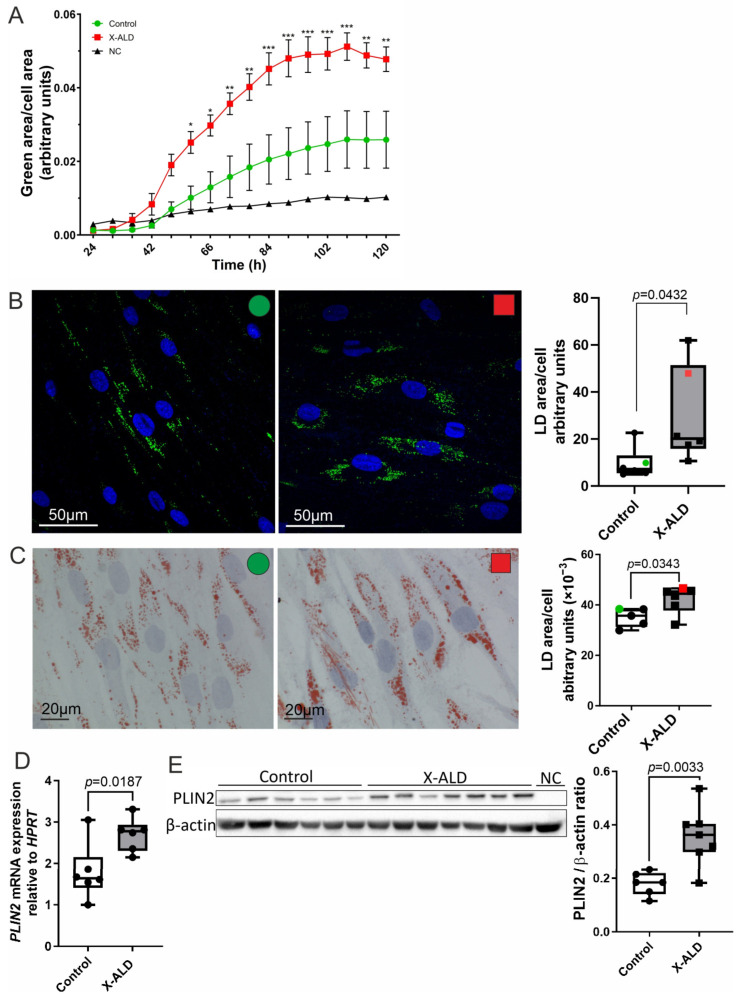
Increased lipid droplet induction upon prolonged cholesterol loading in X-ALD vs. control fibroblasts. (**A**–**E**) Control and X-ALD-derived primary fibroblasts were starved for 72 h in LDM and then incubated for 5 days with 10 or 20 µg/mL cholesterol in LDM. (**A**) The time course of LD formation induced by 20 µg/mL cholesterol treatment was monitored in four cell lines/genotype using the neutral lipid stain, BODIPY™ 493/503, and Incucyte^®^ live-cell imaging. Cells were recorded from 24 h through to 120 h after adding cholesterol. The results are expressed as the fraction green fluorescence area/cell area. NC, normal control fibroblast line in LDM with 2% ethanol and BODIPY™ 493/503. The data are depicted as mean ± SD. Two-way ANOVA with Dunnett’s multiple comparisons test: * *p* < 0.05; ** *p* < 0.01; *** *p* < 0.001. (**B**–**E**) Fibroblasts (n = 6 per genotype) were exposed to 10 µg/mL cholesterol for 5 days before analysis. Representative confocal microscopy pictures (**B**) of control (left panel, green dot) and X-ALD (middle panel, red square) cells stained with BODIPY™ 493/503 (LDs, green fluorescence) and DAPI (nuclei, blue). Light microscopy pictures (**C**) of control and X-ALD cells stained with Oil Red O and haematoxylin. Images in (**B**,**C**) were analysed using ImageJ software and the results were expressed as LD area/cell (right panels). The microscopy views correspond to the colour-coded samples in the graphs. (**D**) Relative Perilipin 2 (PLIN2) mRNA levels, normalised to HPRT1, determined by RT-qPCR. (**E**) PLIN2 protein expression normalised to β-actin was assessed by western blot analysis: Original images could be found in Appendix A—original-images. (**E**). In (**B**–**E**), the box plots show individual data points and the median; statistically significant differences determined using unpaired two-sided Student’s *t*-test are displayed as exact *p*-values.

**Figure 4 biomolecules-13-01333-f004:**
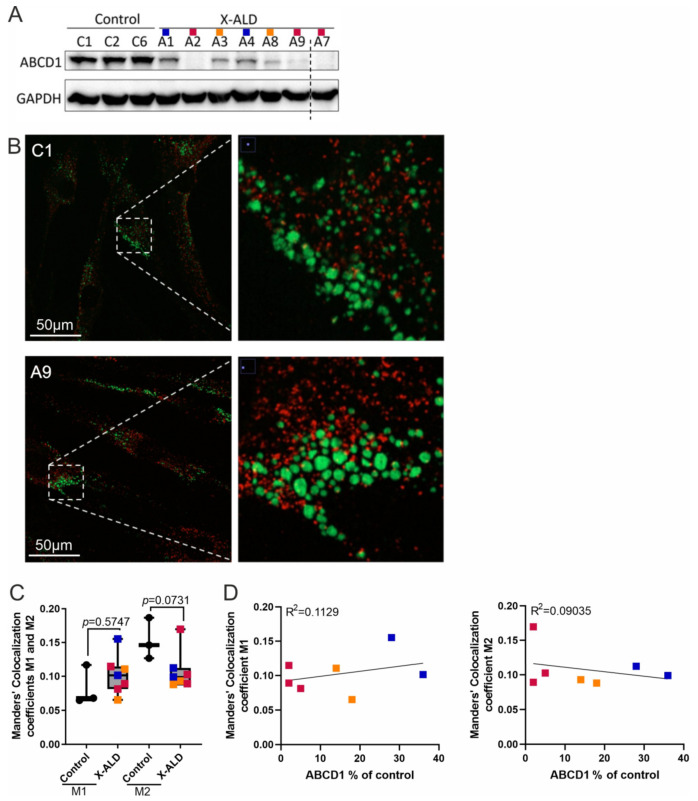
Low interaction between peroxisomes and lipid droplets under cholesterol-loading conditions in both control and X-ALD fibroblasts. (**A**) Western blot analysis depicting various levels of stable ABCD1 protein and GAPDH for normalisation of loading in the cell lines used for the co-localisation study in standard RPMI medium. The samples are colour-coded according to their ABCD1 content (blue, >20%; orange, 5–20%; red, <5%). A cut in the image is indicated by a dashed line; for the full set and quantification of ABCD1 protein levels, see Appendix A. (**B**–**D**) Control (n = 3) and X-ALD-derived fibroblasts (n = 7) were starved in LDM, loaded with 20 µg/mL cholesterol for 72 h and stained with BODIPY™ 493/503 (for LDs) and anti-PMP70 antibody (for peroxisomes). Representative ultra-high resolution confocal images of LDs (green) and peroxisomes (red) and their interactions (yellow) in cholesterol-treated cells (control C1 and X-ALD A9) Original images could be found in Appendix A—original-images (**B**). The extent of interaction between LDs and peroxisomes is expressed as Manders’ co-localisation coefficients, M1 (% of green pixel with red component) and M2 (% of red pixel with green component) (**C**), one-way ANOVA with Sidak’s multiple comparisons test. Linear regression analysis of the relationship between M1 or M2 and the level of mutated ABCD1 protein in X-ALD cells revealed no significant correlation (**D**).

**Figure 5 biomolecules-13-01333-f005:**
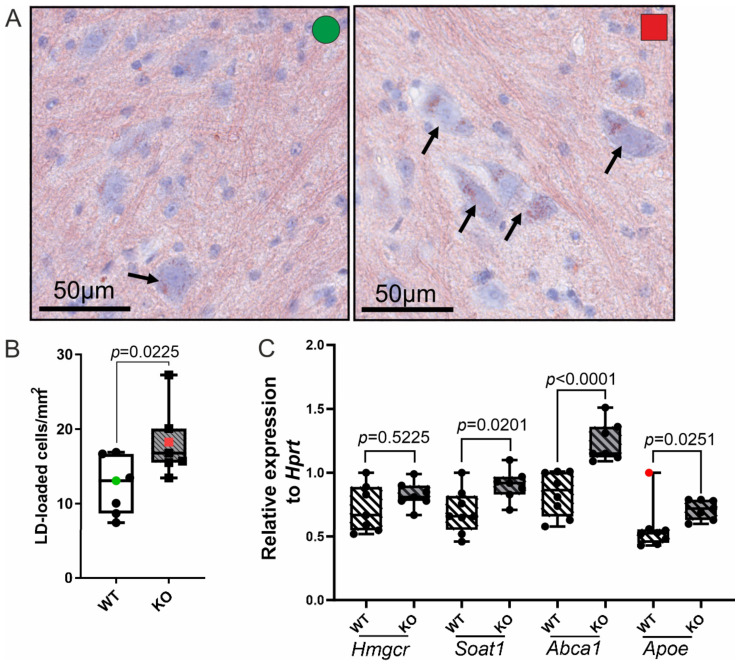
Dysregulation of the cholesterol-related gene expression and increased numbers of LD-positive neurons in the spinal cord Abcd1 KO mice. (**A**–**C**) Spinal cord tissue was dissected from 12-month-old WT or Abcd1 KO mice (n = 7 each) for RNA extraction or cryosectioning and Oil Red O (ORO) staining of LDs. Representative light microscopy views from the ventral horn of ORO-stained WT (green circle) and Abcd1 KO (red square) lumbar spinal cord, counterstained with haematoxylin (blue cell nuclei and, in neurons, rough ER) (**A**) show the somas of several motor neurons with LDs (black arrows). For quantification of neutral lipid accumulation, the number of LD-loaded motor neurons was normalised to the tissue area (ventral horn grey matter) (**B**). The coloured data points correspond to the samples shown in (**A**). The relative mRNA levels of the cholesterol-related genes: Hmgcr (synthesis), Soat1 (esterification), Abca1 and Apoe (export), and for normalisation, Hprt, were determined by RT-qPCR (**C**). The box plots display all individual values and the median. The red dot indicates an extreme outlier (>3 × IQR), excluded from the statistical analysis. The unpaired two-sided Student’s *t*-test in was performed in (**B**) and one-way ANOVA with Sidak’s multiple comparisons test in (**C**).

**Figure 6 biomolecules-13-01333-f006:**
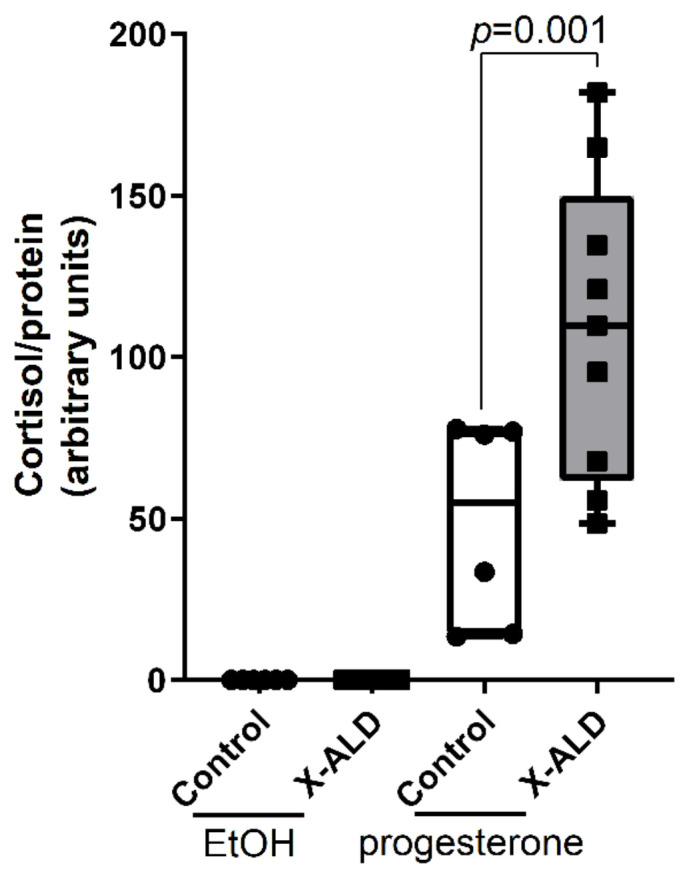
X-ALD fibroblasts release higher amounts of cortisol upon stimulation with progesterone. Control (n = 6) or X-ALD (n = 10) fibroblasts were treated with vehicle (EtOH) or 1 μM progesterone for 24 h. Cortisol levels were measured in the supernatants using Luminex xMAP technology and normalised to the protein content of the harvested cells. The results are presented as box plots showing all values and the medians. Data were analysed by conducting one-way ANOVA with Sidak’s multiple comparisons test.

**Table 1 biomolecules-13-01333-t001:** Description of X-ALD patient-derived fibroblast lines and relative ABCD1 protein levels.

X-ALD Cell Line	ABCD1 Mutation	Protein Alteration	ABCD1 Level (% of Control) †	Phenotype
A1	c.1817C>T	p.Ser606Leu	36	presymptomatic
A2	c.3G>A	p.Met1Ile	<5	AMN
A3	c.310C>T	p.Arg104Cys	18	presymptomatic
A4	c.1817C>T	p.Ser606Leu	28	presymptomatic
A5	c.1814T>C	p.Leu605Pro	5	CALD
A6	N/A	N/A	<5	CALD
A7	N/A	N/A	<5	AMN
A8	c.1451C>G	p.Pro484Arg	14	AMN
A9	c.1907G>T	p.Ser636Ile	5	AMN
A10	c.1817C>T	p.Ser606Leu	47	presymptomatic
A11	N/A	N/A	<5	presymptomatic
A12	c.1679C>T	p.Pro560Leu	<5	CALD
A13	c.1415_16delAG	p.Gln472Argfs*83	<5	AMN
A14	N/A	N/A	85	AMN
A15	c.311G>A	p.Arg104His	30	CALD

† The relative protein level was calculated based on the ABCD1 to GAPDH ratio obtained from western blots (Appendix A) and expressed as a fraction (%) of the ratio of the healthy control C1 (100% ABCD1). N/A, not available.

## Data Availability

All data are presented in the manuscript and/or Appendix A and are available upon request.

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
