# Peer review of "ABCD1 Transporter Deficiency Results in Altered Cholesterol Homeostasis"

_biomolecules, 2023, doi:10.3390/biom13091333_

Round 1
Reviewer 1 Report
ABCD1 transporter deficiency results in altered cholesterol homeostasis
This is an important paper addressing cholesterol metabolism in X-linked adrenoleukodystrophy (ALD) with the studies in 15 ALD fibroblasts with known mutations and phenotype vs control fibroblast and the ALD mouse model, the Abcd1 KO, vs WT mouse.
The interesting and important findings follow.
Lipidomic analyses of fibroblast cultures grown under normal culture conditions showed accumulation of cholesterol and cholesterol esterified with fatty acids, more in ALD fibroblasts than in control. The cholesterol esters showed a wide range of very long chain, saturated, mono as well as polyunsaturated fatty acids with the most prevalent fatty acid, C26:0
In the 8-month-old Abcd1 KO mouse, compared with WT mouse, the increased cholesterol esters, with the most abundant species as cholesterol esterified with C26:0, were found in brain and spinal cord, with more in the spinal cord, possibly reflecting the development of spinal cord disease.
The most amazing findings of the cholesterol ester studies, were the variety of fatty acids found to be esterified with cholesterol in the ALD fibroblasts as well as the Abcd1 mouse model brain and spinal cord., Not only were the very long saturated and monounsaturated fatty acid found, but also there were many polyunsaturated fatty acids esterified and found to be significantly increased when compared to controls.
Note to authors: It would be of interest to list (in the supporting documents) the molecular ions of the cholesterol esters measured by LCMSMS as there are unique species of cholesterol esters identified.
As increased cholesterol is stored in lipid droplets, the observation of increased cellular cholesterol content in fibroblasts was followed up with experiments of lipid droplet, LD, formation after growing cells in lipid depleted media, and adding cholesterol to the media and following LD formation using the fluorescent dye BODIPY. The ALD cells showed higher LD formation than did the control cells.
LD formation was also measured in the spinal cord of the 12-month-old ALD KO mice and compared with WT mice. There was also an increase in the number of LD –loaded motor neurons in the ventral horn of the lumber spinal cord, normalized to the ventral horn area.
The interaction between lipid droplets and peroxisomes was measured and found to be similar between ALD fibroblasts and controls.
The key genes involved in cholesterol homeostasis were studied in fibroblasts. The genes, 3-hydroxy-3-methylglutaryl-coenzyme A reductase (HMGCR), sterol-O-acyltransferase a (SOAT1), neutral cholesterol ester hydrolase 1 (NCEH1) and ATP-binding cassette transporter A1 (ABCA1) were measured by RT-qPCR. Fibroblast cultures of controls and ALD were grown in low cholesterol media, complete media, and high 10ug/ml. cholesterol media There was, as expected an increase in HMGCR when cells were grown in lipid depleted media, but there was no significant difference in HMGCR expression in control and ALD cells grown in high cholesterol media, but there was higher expression of SOAT1, NECH1, and ABCA1 in ALD cells relative to controls when grown in regular media and more change in the high cholesterol media .
Also of great interest was the higher response of cortisol measured after progesterone stimulation in ALD fibroblasts when compared with control cells.
The discussion points of interest:
The rate of cholesterol esterification of VLCFA compared with long chain FA is similar, about 35%; whereas the rate of degradation is only 5%. These findings were published about 50 years ago and retain the interest in studying cholesterol metabolism in ALD today.
The oxysterols are strong activators of LXR and induce ABCA1 and APOE and the desaturase SCD1, perhaps leading to increased levels of mono and polyunsaturated cholesterol esters in ALD.
The top dysregulated pathway in ALD macrophages is cholesterol metabolism.
Thus, there is a need for cholesterol lowering and VLCFA lowering in the therapy for ALD.
Author Response
We would like to thank Reviewer 1 for all insightful comments and acknowledgement of the importance of our work. We appreciate the time and effort invested in reviewing our research paper. In order to address you recommendation:
“Note to authors: It would be of interest to list (in the supporting documents) the molecular ions of the cholesterol esters measured by LCMSMS as there are unique species of cholesterol esters identified.”
Changes in the manuscript:
we have now included tables with the raw data derived from CE-FAs species lipidomic analysis of both human control and X-ALD fibroblasts, along with the lipidomic data of the CNS of WT and Abcd1-KO mice. These tables (Supplementary Table 2, 3 and 4) can be found in the supplementary material. We appreciate this suggestion and firmly believe that it has indeed enhanced the value of the manuscript.

Reviewer 2 Report
The manuscript of Buda et al. examines cholesterol metabolism in primary fibroblasts from patients with X-linked adrenoleukodystrophy (X-ALD). There is some evidence in the literature that shows that disruption of the peroxisomes affects not only the metabolism of very long chain fatty acids (VLCFA), but also cholesterol metabolism. The authors have substantiated their study well. The study was performed well with careful analysis. Overall I found this manuscript to be quite good. While the authors have introduced a very important topic of the role of cholesterol in X-ALD, I found the biggest omission was the involvement of triglyceride(TG). TG and CE often coincide in lipoproteins and lipid droplets (LDs). Certainly, some mention of TG needs to be included. I do have some comments.- I found that the authors didn't quite give enough information about X-ALD. This limits the interpretation of the data. For example, X-ALD causes a peroxisome deficiency. What does that mean? Peroxisomes don't form, or the functions of the peroxisome are deficient? In this case, peroxisomes are incapable of degradation/oxidation of VLCFAs. What types of oxidation can occur in the peroxisomes (alpha, beta or omega)? Which of these are impaired? If they cannot be degraded, where are they stored? Are there any known normal effects/functions of peroxisomes on sterols?
- TGs are often the most common hydrophobic lipid in LDs, but not always, for example in steroidogenic tissues. I found it completely reasonable that the authors focussed their study on CE, but also completely odd that they hardly mentioned TG. Surely some comments need to be stated and possibly some figures sections added to acknowledge the possible role of VLCFAs being stored in TGs.
- cholesterol efflux study: first, in the abstract (line 28), I believe the authors meant to say cholesterol efflux not CE efflux. The authors have chosen to perform the efflux assay using human serum as the acceptor. I find this choice acceptable but no without a caveat. Apolipoprotein A-I is a specific acceptor that directly interacts with ABCA1. HDL is a specific acceptor that interacts with ABCG1 (although not in mouse tissues). Both of these would measure specific cholesterol efflux. In serum, there are many lipoproteins, serum albumin and perhaps other proteins that would serve as non-specific acceptors of cholesterol. When the authors chose human serum, they must state clearly that they are measuring mostly non-specific cholesterol efflux, and this may affect the conclusions. Figure 2F shows what I mean with such a high background of efflux.
- I would have suspected that the vast majority of VLCFAs would form triglycerides (TG) and be present in the LDs. The authors have deliberately chosen not to discuss this, but I think it is worth mentioning. If this is not the case, that the vast majority of VLCFAs form in CE rather than TGs, then the authors needs to state this even more clearly, and then it would substantitate their work even more so. So, please clarify.
- Section 3.2: While HMGCoA-reductase is one main gene, there are others not assessed in this paper. SREBP2 serves as an excellent readout of the cholesterol homeostasis level (not only of the amount of protein, but also whether it has been cleaved to its active nuclear transcription factor form). Apolipoprotein E is an LXRa activatable gene relevant to nervous system tissue. While mRNA measurements are helpful, protein levels are more reliable as HMGCoAreductase is regulated post-translationally
- Line 349: it is unclear to me why SOAT1 would be downregulated under high cholesterol load, when it would be important to prevent free cholesterol accumulation and toxicity.
- Section 3.2: I appreciate the authors' statement that LXR may be regulating the impaired response in X-ALD cells. Use of the LXR agonist (Supp. Figure 2A-C) supports this observation, partly. I would recommend that Supp. Figure 2A-C be moved up into the main figures because of this and supplemented with the use of a profoundly better LXR agonist (GW3965 or T-901317) and inhibitor (geranylgeraniol). This would certainly support the authors conclusions.
- Figure 3.3: I find Figure 3 to be of high quality with the exception of one thing. There is no measurement of TGs. Surely addition of lipids is causing an accumulation of TGs. It would be important to know the relative accumulation of CE and TG. Are these lipid droplets only composed of CE? Are LDs also increased in X-ALD when cells are given oleate loading?
- Section 3.4: Figure 4, While it is an important observation that cholesterol loading does not induce peroxisome:LD interactions, I wonder if cholesterol/fatty acid starvation would promote this interaction?
- It should be noted that NCEH is not the only documented cholesterylester esterase. So, while it is likely that NCEH plays a role and is important, the authors should also acknowledge that other CE esterases exist (HSL, TGH) some with cross-reactivity to TG hydrolysis.
I just found a few minor errors. Overall, no concerns.
Author Response
POINT-BY-POINT RESPONSE Buda et al. “ABCD1 transporter deficiency results in altered cholesterol homeostasis”
Reviewer 2
We thank Reviewer 2 for all the insightful and comprehensive comments and appreciate the time and effort invested in reviewing our research paper. The constructive suggestions and valuable feedback have significantly contributed to the refinement of our manuscript.
We have carefully considered all comments and accordingly added new data, where possible, within the short timeframe given for the revision and/or addressed the concerns in the text. The necessary revisions in the manuscript and our response to each point are outlined below:
- I found that the authors didn't quite give enough information about X-ALD. This limits the interpretation of the data. For example, X-ALD causes a peroxisome deficiency. What does that mean? Peroxisomes don't form, or the functions of the peroxisome are deficient? In this case, peroxisomes are incapable of degradation/oxidation of VLCFAs. What types of oxidation can occur in the peroxisomes (alpha, beta or omega)? Which of these are impaired? If they cannot be degraded, where are they stored? Are there any known normal effects/functions of peroxisomes on sterols?
We thank the Reviewer for this comment. Indeed, we did not provide sufficient information about the metabolic defect in X-ALD.
Changes in the manuscript:
We have now extended the introduction and included the missing information:
“The molecular cause of X-ALD is mutations in the ABCD1 gene encoding a peroxisomal membrane transporter, mediating the transport of CoA activated very long-chain fatty ac-ids (VLCFAs, ≥ C22:0) into the peroxisomal lumen. Although otherwise intact peroxisomes are present in X-ALD, this single-transporter deficiency impairs the import of VLCFAs into these organelles, preventing their subsequent degradation via β-oxidation. Given that peroxisomes are the only organelle capable of VLCFA degradation in mammals, these fatty acids become incorporated into various lipid classes and accumulate in the tissues and body fluids of X-ALD patients, which constitutes a biomarker for the diagnosis of X-ALD [3], [4].”
- TGs are often the most common hydrophobic lipid in LDs, but not always, for example in steroidogenic tissues. I found it completely reasonable that the authors focussed their study on CE, but also completely odd that they hardly mentioned TG. Surely some comments need to be stated and possibly some figures sections added to acknowledge the possible role of VLCFAs being stored in TGs.
We thank the Reviewer for bringing up this important point for further clarification. Our work was focused on the alterations in cholesterol homeostasis in X-ALD, which is particularly important in the context of steroidogenic tissues and in phagocytes removing myelin debris in CALD brain lesions. However, we agree that the importance of TG accumulation in X-ALD should not be neglected. This issue has been addressed in several studies like, for example, Lee et al. 2019 reported TG accumulation in human X-ALD fibroblasts and Fourcade et al. 2020 reported TG accumulation in human X-ALD fibroblasts and the spinal cord of Abcd1 mice, which could be restored by the high biotin treatment.
Our lipidomic analysis in human fibroblasts also revealed TG accumulation in X-ALD with a similar pattern as observed by Lee et al.
Changes in the manuscript:
We now included an additional figure, new Supplementary Figure 2, showing a volcano plot of all significantly different TG(FA) species (A) and the top three significantly altered TG(FA)s (B) in X-ALD vs control fibroblasts.
We also included more information about the accumulation of TG in X-ALD in the main text, in Results section 3.1:
“It is worth noting that VLCFAs accumulate in various lipid species and to a significant degree also in triglycerides (TG). Our lipidomic analysis revealed the accumulation of TG(FA)s in X-ALD fibroblasts vs control fibroblasts (Supplementary Figure 2A, B) with a trend that aligns with findings from other lipidomic studies on X-ALD fibroblasts [43] and spinal cord tissue from Abcd1 KO vs WT mice [44]. We found a highly significant increase for TG(60:2), TG(62:2) and TG(62:3). Although we only have the sum of TG sn-1-3, based on fatty acid abundancy, all of them are highly likely to contain a VLCFA”
And in the discussion:
“While acknowledging the importance of TG accumulation in the context of X-ALD, here we focused on investigating potential disruptions in cholesterol homeostasis stemming from the ABCD1 defect, which is of great importance for steroidogenic tissues or phagocytes removing myelin debris in CALD brain lesions.”
- cholesterol efflux study: first, in the abstract (line 28), I believe the authors meant to say cholesterol efflux not CE efflux. The authors have chosen to perform the efflux assay using human serum as the acceptor. I find this choice acceptable but no without a caveat. Apolipoprotein A-I is a specific acceptor that directly interacts with ABCA1. HDL is a specific acceptor that interacts with ABCG1 (although not in mouse tissues). Both of these would measure specific cholesterol efflux. In serum, there are many lipoproteins, serum albumin and perhaps other proteins that would serve as non-specific acceptors of cholesterol. When the authors chose human serum, they must state clearly that they are measuring mostly non-specific cholesterol efflux, and this may affect the conclusions. Figure 2F shows what I mean with such a high background of efflux
We thank the Reviewer for finding the error in the Abstract. Of course, we meant cholesterol efflux, not CE efflux and corrected the mistake. Moreover, we completely agree with the comment regarding choosing human serum as a cholesterol acceptor.
Changes in the manuscript:
We now included a comment in the Discussion about serum being a non-specific cholesterol acceptor:
“Distinct acceptors are specific for different cholesterol efflux pathways. To measure specific cholesterol efflux, ApoA-1 is used to reflect ABCA1-dependent cholesterol efflux, while HDL is selective for ABCG1 or SR-B1-dependent cholesterol efflux. In our study, we used human serum as a cholesterol acceptor that contains a range of lipoproteins beyond ApoA-1, as well as albumin, which can bind cholesterol non-specifically and, thus, measure the combined transporter-specific and non-specific cholesterol efflux.”
- I would have suspected that the vast majority of VLCFAs would form triglycerides (TG) and be present in the LDs. The authors have deliberately chosen not to discuss this, but I think it is worth mentioning. If this is not the case, that the vast majority of VLCFAs form in CE rather than TGs, then the authors needs to state this even more clearly, and then it would substantiate their work even more so. So, please clarify.
We are grateful for the Reviewer for bringing this up. Under the standard cell culture conditions, primary fibroblasts contain only a small amount of LDs, which of course can comprise TG as well as CE. This was the reason why, for most of our experiments, we first lipid-starved the fibroblasts and then loaded them with cholesterol in order to induce CE-enriched LDs. Regarding LD staining in the murine spinal cord, based on the presented data, we can, indeed, not determine the fraction of VLCFA-containing cholesteryl esters or triglycerides in the lipid droplets. This information was certainly missing.
Changes in the manuscript:
We added it in the Results section 3.5:
“It is important to acknowledge that since also TGs accumulate in the spinal cord of X-ALD mice [44], we cannot determine the extent of CE and TG in the LDs.”
- Section 3.2: While HMGCoA-reductase is one main gene, there are others not assessed in this paper. SREBP2 serves as an excellent readout of the cholesterol homeostasis level (not only of the amount of protein, but also whether it has been cleaved to its active nuclear transcription factor form). Apolipoprotein E is an LXRa activatable gene relevant to nervous system tissue. While mRNA measurements are helpful, protein levels are more reliable as HMGCoAreductase is regulated post-translationally
We thank the Reviewer for this insightful comment. We agree that the expression of several key genes in cholesterol homeostasis are subject to post-translational regulation; and it would indeed be valuable to assess their protein expression and modifications to obtain deeper insights and further substantiate our findings. We sincerely regret that due to time constraints, we are unable to present new data concerning the protein expression of cholesterol-related genes. However, what we can offer are comparisons of SREBP2 and LDLR mRNA levels between control and X-ALD fibroblasts under all three culture conditions. The expression patterns of these genes exhibit striking similarities to that of HMGCR. To provide this information, we have incorporated these results in the text and into a new Supplementary Figure 3A and B.
Changes in the manuscript in (Results section 3.2):
“Furthermore, the expression patterns of sterol regulatory element-binding protein 2 (SREBP2), encoding a key regulator of HMGCR and other genes related to cholesterol biosynthesis, along with the low-density lipoprotein receptor (LDLR), which is likewise regulated by SREBP2 [46], were strikingly similar to that observed for HMGCR (Supplementary Figure 3 A and B)..”
We acknowledge the significance of Apolipoprotein E as a crucial cholesterol acceptor in the context of nervous system tissue. While we assumed that APOE expression might not hold substantial relevance in our in vitro system, we did not assess the expression of this gene in our fibroblast culture model. Nonetheless, we evaluated the expression of Apoe in the spinal cord of Abcd1 KO vs WT mice and incorporated these findings into Figure 5C.
Changes in the manuscript (Results section 3.5):
“To investigate cholesterol homeostasis in the spinal cord of Abcd1 KO mice in more detail, we analysed the expression of the cholesterol-related genes Hmgcr, Soat1, Abca1 and Apoe in the spinal cord of 12-month-old WT and Abcd1 KO mice (Figure 5C). Similar to our findings in human X-ALD fibroblasts, RT-qPCR revealed no genotype difference in the mRNA levels of Hmgcr. In contrast, Soat1, Abca1 and Apoe expression were significantly higher in the KO animals, reinforcing the results obtained in vitro from X-ALD fibroblasts (Figure 2C, E).”
We also added the paragraph about the importance of ApoE in the nervous system in the Discussion:
“APOE serves as the primary extracellular transporter of lipids and cholesterol, predomi-nantly secreted by astrocytes. The APOE ε4 allele is the primary genetic risk factor for late-onset Alzheimer's disease [61]. Furthermore, a recent observation revealed that male X-ALD patients with the APOE4 genotype exhibited increased cerebral involvement, as indicated by the severity assessed through MRI [62]. This makes our observation of Apoe up-regulation in the spinal cord of Abcd1 KO mice intriguing, possibly suggesting an increased need for cholesterol export in X-ALD.”
- Line 349: it is unclear to me why SOAT1 would be downregulated under high cholesterol load, when it would be important to prevent free cholesterol accumulation and toxicity.
In the literature, SOAT1 (ACAT1) expression is described to be unchanged in various cell types under high-cholesterol treatment as well as with 25-HC in contrast to the small intestine- and liver-selective SOAT2 (ACAT2) (Wang et al. 2017), explaining why in liver or small intestine the activity is increased under conditions of high cholesterol for efficient dietary cholesterol absorption and lipoprotein assembly. In our study, we used a long-term, high-cholesterol treatment paradigm that might additionally be in part responsible for the observed downregulation of SOAT1.
Changes in the manuscript:
To make this point clearer, we have added the “long-term”:
“Whereas control cells were able to downregulate SOAT1 expression under long-term high-cholesterol load (p=0.0029), …”
- Section 3.2: I appreciate the authors' statement that LXR may be regulating the impaired response in X-ALD cells. Use of the LXR agonist (Supp. Figure 2A-C) supports this observation, partly. I would recommend that Supp. Figure 2A-C be moved up into the main figures because of this and supplemented with the use of a profoundly better LXR agonist (GW3965 or T-901317) and inhibitor (geranylgeraniol). This would certainly support the authors conclusions.
We appreciate the Reviewer’s recommendations regarding usage of more potent LXR agonists as well as antagonists to substantiate our findings. However, we believe that adding much more potent agonists might elicit a much higher response in both control and X-ALD cells potentially without introducing novel insights to our results. Moreover 25HC have been reported to be elevated in X-ALD cells, therefore we believe that our choice was appropriate. We sincerely apologise that because of the time constrains, we are unable to conduct an experiment using an LXR inhibitor, however we think it is worth to explore in the future. Since we only hypothesise about the involvement of LXR pathway in our observation and do not intend to make a definite statement, we opted to retain the Supplementary Figure 2A-C (now Supplementary Figure 4A-C) within the supplementary information. Nonetheless, we agree with the Reviewer’s viewpoint that delving deeper into the LXR pathway in the context of X-ALD is indeed worth further investigation.
Later on, we discuss the impact of TO901317 on VLCFA and CE-VLCFA levels in the mouse CNS. Please refer to point number 10 for more details.
- Figure 3.3: I find Figure 3 to be of high quality with the exception of one thing. There is no measurement of TGs. Surely addition of lipids is causing an accumulation of TGs. It would be important to know the relative accumulation of CE and TG. Are these lipid droplets only composed of CE? Are LDs also increased in X-ALD when cells are given oleate loading?
We appreciate the Reviewer's input. We partially explained this issue in the point 4. In our experiment, we first deprived the fibroblasts of lipids (using lipid-depleted medium), then introduced cholesterol into the lipid-depleted medium to stimulate the formation of cholesteryl ester-enriched lipid droplets. Because we did not analyse the lipid composition of those LDs, we only can assume that they predominantly contain CE. Moreover, previously we subjected human fibroblasts to treatment with OA, revealing markedly distinct dynamics, in which OA triggered a considerably faster induction of LDs compared to cholesterol. However, the results obtained from these experiments did not provide definitive insights into the comparison between control and X-ALD fibroblasts. While addressing this aspect is of significant interest for future investigations, our primary aim for the present research paper was focussed on the cholesterol-related aspects.
- Section 3.4: Figure 4, While it is an important observation that cholesterol loading does not induce peroxisome:LD interactions, I wonder if cholesterol/fatty acid starvation would promote this interaction?
We appreciate the Reviewer addressing this issue. Indeed, LD-peroxisome interaction is induced by starvation in mouse epididymal white adipose tissue (eWAT), as indicated by Kong et al. 2020. In our experiment, we aimed to evaluate LD-peroxisome interaction under the cholesterol-loading conditions. However, we agree that it would be of great interest to repeat this experiment and asses the interaction in the lipid starvation conditions. In this case LDs still would have to be induced, as in the standard condition there is only a low amount of residual LDs. We have taken this valuable point into consideration and integrated it into our Discussion.
Changes in the manuscript:
“Given that fasting-induced lipolysis, triggered by activating adipose triglyceride lipase (ATGL), can increase LD-peroxisome interaction [70], it is plausible that analogous mechanisms might come into play concerning LDs containing cholesteryl esters (CE). Consequently, conducting an assessment of LD-peroxisome interactions under lipiddepleted conditions, after inducing LDs with cholesterol, holds significant potential and deserves further investigation.”
- It should be noted that NCEH is not the only documented cholesterylester esterase. So, while it is likely that NCEH plays a role and is important, the authors should also acknowledge that other CE esterases exist (HSL, TGH) some with cross-reactivity to TG hydrolysis.
We greatly appreciate the Reviewer raising another crucial aspect. The Reviewer's observation is entirely accurate; there are indeed several cholesterylester esterases, as well as other lipases, with overlapping substrate specificities. However, their relevance varies depending on specific tissues. For instance, HSL (encoded by LIPE) primarily plays a role in adipose tissue. On the other hand, carboxyl ester lipase (CEL) is an enzyme produced by the adult pancreas and possesses the capability to hydrolyse cholesteryl esters, vitamin esters, triacylglycerol, phospholipids, and lysophospholipids. NCEH1 is expressed in multiple tissues, including macrophages and cells of the central nervous system. According to the existing literature, it exhibits specificity for cholesteryl esters and holds significance for overall cholesterol metabolism. This is why we chose to focus on NCEH1 in our research related to cholesterol. Nonetheless, it is true that we cannot rule out the potential involvement of other lipases in the hydrolysis of cholesteryl esters in fibroblasts as well as the murine spinal cord.
In the old studies of CE hydrolase activity in X-ALD that we cite in the Discussion (Tadashi Ogino and Suzuki 1981) and (T Ogino et al. 1978) brain homogenates were used, therefore including all lipases present in the tissue.
To further target this point, we reinvestigated the CE(26:0) and CE(26:1) levels in the murine CNS tissue, where a reduction of VLCFA was observed after treatment of control and X-ALD mice with LXR agonist TO901317 for 10 weeks from a recent study (Raas et al. 2021). Interestingly both investigated CEs did not change neither in spinal cord nor in the brain in spite of the reduction of overall VLCFAs.
Thus we have added a new Supplementary Figure 5 to the manuscript
Changes in the manuscript:
We have added in the Result section 3.2:
“In a previous study, dietary treatment of X-ALD mice with the LXR agonist TO901317 was able to reduce total levels of VLCFAs in the brain and spinal cord [32]. Therefore, we reanalysed the data from that study to determine whether CE(26:0) and CE(26:1) were also reduced. Neither the levels of CE(26:0) nor CE(26:1) were altered after TO treatment (Supplementary Figure 5).
Together, these data indicate that in X-ALD the response to cholesterol exposure is altered, probably to counteract further increases in toxic, free cholesterol as well as CEs, which may accumulate due to reduced CE-VLCFA hydrolysis.”
We have added in the Discussion:
“The advantage of these studies in brain or fibroblast homogenates is that the activities of all CE-synthesising and CE-hydrolysing enzymes are reflected including SOAT1 and NCEH1. Our observation, that TO901317 treatment able to reduce total VLCFA in the CNS [32] but not VLCFAs in the CE fraction (Supplementary Figure 5), is in good agreement with the hypothesis that the hydrolysis of CE-VLCFA substantially contribute to the VLCFA accumulation in the CE fraction.”
Bibliography:
Fourcade, Stéphane, Leire Goicoechea, Janani Parameswaran, Agatha Schlüter, Nathalie Launay, Montserrat Ruiz, Alexandre Seyer, et al. 2020. High-Dose Biotin Restores Redox Balance, Energy and Lipid Homeostasis, and Axonal Health in a Model of Adrenoleukodystrophy. Brain Pathology. Vol. 30. https://doi.org/10.1111/bpa.12869.
Kim, Jungsu, Jacob M. Basak, and David M. Holtzman. 2009. “The Role of Apolipoprotein E in Alzheimer’s Disease.” Neuron 63 (3): 287–303. https://doi.org/10.1016/j.neuron.2009.06.026.
Kong, Jinuk, Yul Ji, Yong Geun Jeon, Ji Seul Han, Kyung Hee Han, Jung Hyun Lee, Gung Lee, et al. 2020. “Spatiotemporal Contact between Peroxisomes and Lipid Droplets Regulates Fasting-Induced Lipolysis via PEX5.” Nature Communications 11 (1): 1–16. https://doi.org/10.1038/s41467-019-14176-0.
Lee, Dong Kyu, Nguyen Phuoc Long, Juwon Jung, Tae Joon Kim, Euiyeon Na, Yun Pyo Kang, Sung Won Kwon, and Jiho Jang. 2019. “Integrative Lipidomic and Transcriptomic Analysis of X-Linked Adrenoleukodystrophy Reveals Distinct Lipidome Signatures between Adrenomyeloneuropathy and Childhood Cerebral Adrenoleukodystrophy.” Biochemical and Biophysical Research Communications 508 (2): 563–69. https://doi.org/10.1016/j.bbrc.2018.11.123.
Luo, Jie, Hongyuan Yang, and Bao Liang Song. 2020. “Mechanisms and Regulation of Cholesterol Homeostasis.” Nature Reviews Molecular Cell Biology 21 (4): 225–45. https://doi.org/10.1038/s41580-019-0190-7.
Ogino, T, H H Schaumburg, K Suzuki, Y Kishimoto, and A E Moser. 1978. “Metabolic Studies of Adrenoleukodystrophy.” Advances in Experimental Medicine and Biology 100: 601–19. https://doi.org/10.1007/978-1-4684-2514-7_44.
Ogino, Tadashi, and Kunihiko Suzuki. 1981. “Specificities of Human and Rat Brain Enzymes of Cholesterol Ester Metabolism Toward Very Long Chain Fatty Acids: Implication for Biochemical Pathogenesis of Adrenoleukodystrophy.” Journal of Neurochemistry 36 (2): 776–79. https://doi.org/10.1111/j.1471-4159.1981.tb01657.x.
Orchard, Paul J., Todd W. Markowski, Lee Ann Higgins, Gerald V. Raymond, David R. Nascene, Weston P. Miller, Elizabeth I. Pierpont, and Troy C. Lund. 2019. “Association between APOE4 and Biomarkers in Cerebral Adrenoleukodystrophy.” Scientific Reports 9 (1): 1–10. https://doi.org/10.1038/s41598-019-44140-3.
Raas, Quentin, Malu Clair Van De Beek, Sonja Forss-Petter, Inge M.E. Dijkstra, Abigail Deschiffart, Briana C. Freshner, Tamara J. Stevenson, et al. 2021. “Metabolic Rerouting via SCD1 Induction Impacts X-Linked Adrenoleukodystrophy.” Journal of Clinical Investigation 131 (8). https://doi.org/10.1172/JCI142500.
Wang, Yong-Jian, Yan Bian, Jie Luo, Ming Lu, Ying Xiong, Shu-Yuan Guo, Hui-Yong Yin, et al. 2017. “Cholesterol and Fatty Acids Regulate Cysteine Ubiquitylation of ACAT2 through Competitive Oxidation.” Nature Cell Biology 19 (7): 808–19. https://doi.org/10.1038/ncb3551.

Round 2
Reviewer 2 Report
The authors have addressed all of my concerns sincerely. I have no further comments but to say this is a wonderful manuscript.